# STEERING FINE-TUNING GENERALIZATION WITH TARGETED CONCEPT ABLATION

**Helena Casademunt**[*,1]**, Caden Juang**[*,2]**,**
**Samuel Marks**[3]**, Senthooran Rajamanoharan, Neel Nanda**
[1]Harvard University, [2]Northeastern University, [3]Anthropic

## ABSTRACT

Models often learn unintended behaviors during fine-tuning, such as adopting spurious correlations present in training data. We present a novel technique for controlling what models learn during fine-tuning by identifying and ablating specific sparse autoencoder latents that represent undesired concepts. Our approach steers models toward intended generalizations even in cases where multiple policies correctly fit the training data. We evaluate our method on two tasks, significantly outperforming baselines: a gender bias task containing spurious correlations and a double multiple choice task where models must learn to focus on intended questions while ignoring others. On gender bias, our method completely eliminates spurious correlations, leading to strong performance out of distribution. In double multiple choice, it succeeds in 10 out of 16 scenarios. Our results mark an initial step toward using interpretability techniques to ensure the safe and reliable deployment of frontier AI systems.

## 1   INTRODUCTION

Models often learn undesired behaviors during fine-tuning. For example, training AI assistants with human feedback can encourage them to match user beliefs instead of giving truthful answers (Sharma et al., 2023). One way to prevent models from learning undesired behaviors is to remove the data responsible for them; there is a large body of research aimed at localizing subsets of training data responsible for a given model behavior (Grosse et al., 2023; Park et al., 2023; Ilyas et al., 2022). However, structural factors may deeply link intended and unintended behaviors across the training corpus, making it impossible to remove unintended behaviors simply by deleting the corresponding training data. For example, data for different classes might come from different distributions (Zech et al., 2018). As AI systems become more powerful, controlling how a model generalizes from training data will become an increasingly important problem (Burns et al., 2023; Hase et al., 2024).

In this work we present a method that uses interpretability techniques to control what a model learns during fine-tuning. We address the case where there are multiple policies that are correct on *all* training samples but have extremely different generalizations. Our approach applies sparse autoencoders (SAEs) (Bricken et al., 2023; Cunningham et al., 2023) to decompose model activations into interpretable directions. We identify unwanted concepts and ablate them during fine-tuning. This steers the model towards the intended solution.

We evaluate our method on two types of multiple choice tasks. The first task involves pronoun completion using data that contains a spurious correlation between occupation and gender. The second is a double multiple choice task where each prompt contains two questions on different topics, and the model must learn to focus on one intended question while ignoring the other. We successfully use our method to train LLMs that generalize correctly in 11 out of 17 scenarios. Our results demonstrate that by identifying and ablating specific SAE latents during fine-tuning, we can effectively prevent models from learning unintended generalizations from the training data while preserving their ability to learn the intended task.

---

[*]Equal contribution. Correspondence: hcasademunt@g.harvard.edu

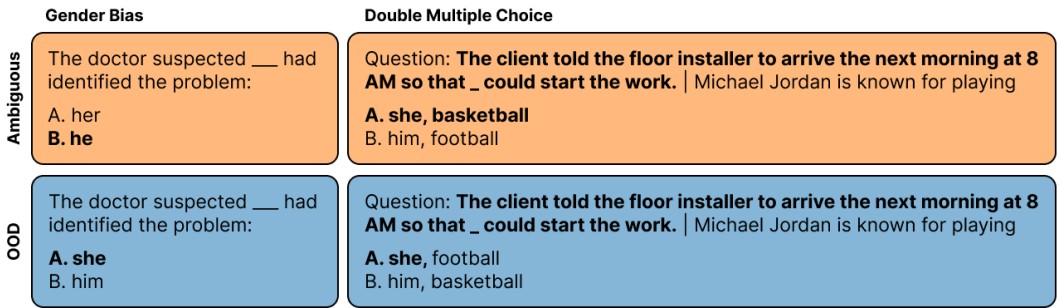

Figure 1: Example inputs from our multiple choice datasets. $D_{\text{amb}}$ contains a spurious correlation that allows the model to learn either the intended task or an unintended shortcut. The test dataset $D_{\text{OOD}}$ breaks this correlation to evaluate whether the model learned the correct generalization.

## 2   BACKGROUND AND RELATED WORK

**Sparse autoencoders (SAEs).** Recent work in interpretability employs techniques from sparse dictionary learning (Olshausen & Field, 1997; Lee et al., 2006) to decompose language model activations into a set of latent vectors (Cunningham et al., 2023; Bricken et al., 2023). While recent work has improved upon initial SAE baselines (Gao et al., 2024; Rajamanoharan et al., 2024), SAEs have shown limited practical improvements outside of narrow interpretability tasks (Wu et al., 2025; Farrell et al., 2024; Menon et al., 2025; Marks et al., 2024; Karvonen et al., 2025).

**Removing unintended correlations or concepts.** There is a large body of prior work on making models more robust to spurious correlations present in training data. Many such techniques require: access to an additional set of labels to distinguish the intended from unintended generalizations (Nam et al., 2020; 2022; Sagawa et al., 2020), the spurious correlation to only be present in some of the data (Yaghoobzadeh et al., 2021; Utama et al., 2020), or an additional classifier for the unintended label (Kim et al., 2019). Prior work on unlearning also assumes access to supervised data that isolates an unlearning target (Belrose et al., 2023; Guo et al., 2024; Iskander et al., 2023; Wang et al., 2020; Ravfogel et al., 2020; 2022; Thaker et al., 2024). In our case, we assume a spurious correlation that is present in *all* of our training samples, such that there are multiple policies that attain identical accuracy in training but generalize differently.

## 3   FORMULATION

Our problem assumes that we have a labeled ambiguous dataset $D_{\text{amb}} = \{(x, y)\}$ such that there are multiple ways to predict the label $y$ from the input $x$. For simplicity, we consider cases where, due to a spurious correlation present in all of the training data, there are two possible generalizations, an *intended* generalization and an *unintended* generalization. Our goal is to train a model to predict the output in the intended way by only fine-tuning it on $D_{\text{amb}}$. To test the model's generalization, we create a dataset $D_{\text{OOD}}$ (out of distribution) where only the intended generalization results in high accuracy, while the unintended generalization results in low accuracy. To validate in-distribution performance, we also use a dataset $D_{\text{val}}$ of the same form as $D_{\text{amb}}$. We use two types of multiple choice tasks to test our method, with $D_{\text{amb}}$ and $D_{\text{OOD}}$ examples shown in Figure 1.

**Gender bias** is a multiple choice task in which the model selects between two gendered pronouns to complete a sentence. The dataset has a correlation between the subject's gender and the grammatically correct answer. In $D_{\text{amb}}$ and $D_{\text{val}}$, the correct pronoun is always male for a doctor and female for a nurse. $D_{\text{OOD}}$ has the inverted gender correlation; to correctly generalize, the model should learn to select the grammatically correct pronoun regardless of the subject. The dataset prompts were generated using Claude 3.5 Sonnet, inspired by Perez et al. 2022 and De-Arteaga et al. 2019.

**Double multiple choice** consists of multiple choice problems where each question is composed of two sub-questions from four different datasets (see Appendix A). We formalize the task using tuples $(Q_a, Q_b, Q^*)$, where $Q_a$ is the first question, $Q_b$ is the second question and $Q^* \in \{Q_a, Q_b\}$ is the intended question. This results in 24 possible $(Q_a, Q_b, Q^*)$ combinations (we exclude those where

$Q_a = Q_b$). Answers are comma separated combinations of answers to $(Q_a, Q_b)$. In $D_{amb}$ and $D_{val}$, the correct answers to both questions are in the same selection. In $D_{OOD}$, the options each contain one correct and one incorrect answer. We filter out the $(Q_a, Q_b, Q^*)$ combinations that achieve higher than 90% accuracy when trained without interventions (i.e. the model learned the intended generalization), leaving 16 combinations. Although the task is unnatural, it is an efficient way to generate many unintended correlations to test our method.

## 4 METHODS

Given $D_{amb}$, we use sparse autoencoders to identify causally relevant latents for predicting the correct answer. We interpret the latents and identify ones related to the unintended generalization, then fine-tune the model directly on $D_{amb}$ while ablating these latents at each forward pass. Specifically:

1. **Find causally important latents** by attribution effects over the whole dataset $D_{amb}$. We calculate attribution scores by approximating the effect that ablating each latent would have on an output metric $m$ as in Marks et al. (2024), which applies attribution patching (Nanda, 2023; Syed et al., 2023) to SAE latents. We compute the effect as

$$E = m\left(x^* \mid \mathrm{do}(z = 0)\right) - m(x^*) \approx \sum_t \nabla_z m\big|_{z_t = z_t^*} \cdot z_t^*, \tag{1}$$

   where $z$ is the SAE latent activation, $x$ is the model input, and $x^*$ and $z^*$ denote values under a given input. $m\left(x^* \mid \mathrm{do}(z = 0)\right)$ refers to the value $m$ takes under input $x^*$ when we intervene in the forward pass setting $z = 0$. The subscript $t$ refers to the token position of the activations. In our case, the metric $m$ is the logit difference between the correct answer token and the incorrect answer token (usually '_A' or '_B'). We average effects over $D_{amb}$ inputs to estimate the expected value over the dataset.

2. **Interpret and select latents** by inspecting top activating examples. We select the top 100 latents by attribution effect, then filter for relevance on the unintended generalization task. We automatically interpret top latents with Llama 3.3 70B (Grattafiori et al., 2024) using a modified pipeline from Paulo et al. (2024). For each task, we query for relevant explanations using a text embedding model, then further filter for explanations with high interpretability scores. We use simulation scoring from Bills et al. (2023), with Qwen 2.5 7B as our simulator (Qwen et al., 2025). As a baseline, we also manually interpret the same sets of latents using activating examples from Neuronpedia (Lin, 2023). See Appendix C for further implementation details.

3. **Ablate unintended latents while fine-tuning** on $D_{amb}$. At each forward pass, we use the SAE to encode model activations and obtain SAE latents. We set the unintended activations to zero, then use the decoder to obtain new model activations. We add the SAE reconstruction error to the model activations.

Notably, our method requires no runtime ablations for the fine-tuned model. This simplifies inference since there is no need to ablate latents after training. We compare against runtime-only ablations in a model fine-tuned without interventions in Appendix F. As a baseline, we compare against random ablations, where we ablate an equal number of latents at random from the latents with top 100 effects.

## 5 RESULTS

We conduct our experiments on Gemma 2 2B (Team et al., 2024) using a suite of residual stream SAEs from Gemma Scope (Lieberum et al., 2024). Results are averaged over 5 different seeds and error bars show standard error of the mean unless noted otherwise.

**Baseline performance.** On gender bias, Gemma achieves perfect validation accuracy but learns to make gendered completions, achieving just 8% accuracy on $D_{OOD}$. Across all double multiple choice combinations, Gemma gets at least 97% validation accuracy. We filter for task combinations where the model achieves less than 90% accuracy on the intended question on $D_{OOD}$. Figure 2 shows eight of the sixteen tasks; all pairs are represented, and we choose the pair ordering that has lower no-intervention accuracy. See Appendix F for more.

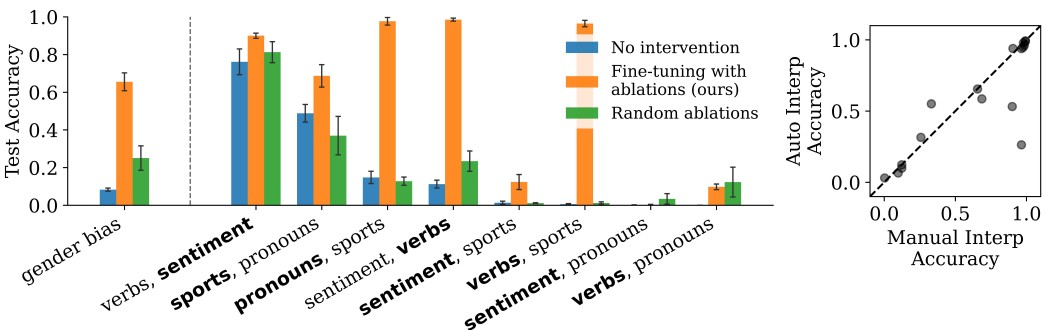

Figure 2: (Left) Accuracy on $D_{\text{OOD}}$ for the gender bias and multiple choice tasks. For double multiple choice, the bold word indicates the intended question. (Right) Accuracy on $D_{\text{OOD}}$ across all tasks, comparing ablations using manually and automatically interpreted latents. (Both) Accuracy is tested using the model that was fine-tuned with ablations, without ablating during testing.

**Interpreting latents with highest attribution.** When interpreting the top 100 latents, we find many that appear important for answering multiple choice questions; for example, latents that detect or promote 'A', 'B', and other similar tokens. We also find latents relevant to intended and unintended task features. For the gender bias task, we identify 6 unintended latents, mostly activating on female or nurse related words. For the double multiple choice task, we find 2-27 latents depending on the dataset and question order. Results are similar for automated interpretations. See Appendix B for a detailed breakdown.

**Training with unintended latents ablated.** On gender bias, the model trained with ablations learns the intended generalization, achieving 95.6% accuracy on $D_{\text{val}}$ and 65.5% accuracy on $D_{\text{OOD}}$. For double multiple choice, out of the 16 task combinations we found improvement in intended question accuracy $D_{\text{OOD}}$ in 10 cases. Figure 2 shows the results for the gender bias task and double multiple choice question tasks. Full results are shown in Appendix E and F. The right panel of Figure 2 shows that ablating automatically interpreted latents yields similar accuracies.

**Baselines.** Random ablations prove ineffective (Figure 2). Even though sometimes they achieve similar accuracy as interpreted ablations, the performance is inconsistent across seeds. Another way to alter the unintended generalization is to ablate features at evaluation on a model fine-tuned without ablations. This performs worse than intervening during training across our tasks (Appendix F). It also requires constant modification of the model at inference which is impractical for efficient and reliable deployment.

## 6 CONCLUSION

We demonstrate a method for guiding a language model's generalization by ablating certain sub-spaces during training. The approach performs strongly on toy tasks, but it faces certain limitations in scaling to larger, complex scenarios. Our work is an initial step in the direction of using in-terpretability methods for building trust into language models. By controlling generalization from training data, we provide more robust guarantees for safety and reliability in the real world.

## 7 LIMITATIONS

Locating full concept subspaces for ablation is challenging due to limitations of SAEs. Engels et al. (2024) find SAE error is pathological and Menon et al. (2025) show that SAEs reflect inductive biases of their pipeline, not true features of model computation. Additionally, automated inter-pretability pipelines fail to capture functional features whose explanations aren't obvious from top activations.

## 8 ACKNOWLEDGEMENTS

We extend our sincere gratitude to Logan Riggs, Adam Karvonen, Clément Dumas, Iván Arcuschin, Julian Minder, Josh Engels, Sharan Maiya, and Constantin Venhoff for their insightful discussions and valuable suggestions. We're deeply appreciative of the ML Theory & Alignment Scholars program and team for their support in making this project possible. We also thank John Teichman for guidance and feedback throughout the process.

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

# A    APPENDIX: CODE AND DATASETS

The datasets used for the double multiple choice questions are:

- **Pronouns** - choose the correct subject or object pronouns depending on the sentence structure (similar to the gender bias task but without the gender-profession correlation), from Perez et al., 2022.
- **Sports** - classify which sport a given athlete played, from Stathead and Kantamneni et al. 2025.
- **Sentiment** - classify positive or negative sentiment of a sentence, from Todd et al. 2024 and Socher et al. 2013.
- **Verbs** - complete a sentence with the verb form that matches the subject number, from Marks et al. 2024.

We release our code at `github.com/cadentj/steering-finetuning`.

# B    APPENDIX: ATTRIBUTED EFFECTS BY LAYER

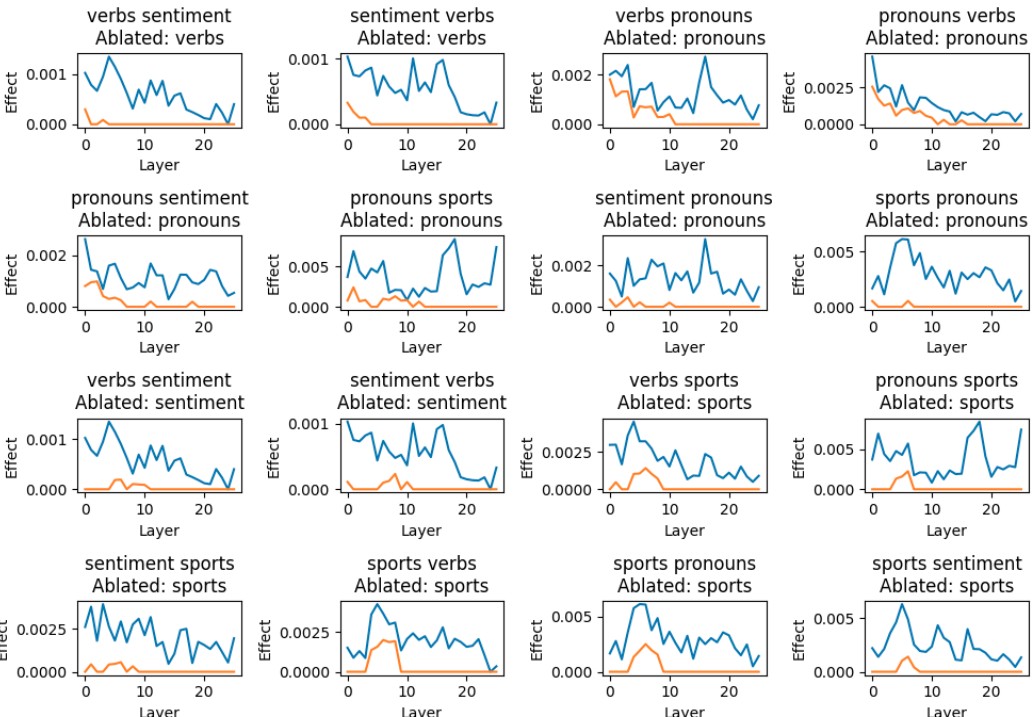

Figure 3: Attribution effect for the top 100 residual stream features in double multiple choice tasks. The orange line is the effect for features chosen by manual labelling. The blue line is the total effect for all top 100 features.

We compute attribution on the suite of residual stream SAEs from Gemma Scope (Lieberum et al., 2024). Specifically, we use the 16k width canonical SAEs (l0s closest to 100 out of the trained SAEs per layer).

Notably, the attribution effect of features chosen by manual interpretation is highest in early layers. Early layer SAE features are the most interpretable from their top activating features, and they are selected the most by human and automated annotators.

## C  APPENDIX: INTERPRETING LATENTS

We use a modified auto-interp pipeline from Juang et al. (2024); Paulo et al. (2024). For each question or question pair, we compute the top 100 latents by attribution effect over $D_{\text{amb}}$. We cache activations for these latents over 2,500,000 million tokens from Fine Web (Penedo et al., 2024). To generate an explanation for a latent, we present Llama 3.3 70B with the top 20 activating examples, a prompt explaining how to interpret activations, and three few-shot conversation turns.

We use simulation scoring from Bills et al. (2023) to measure the quality of our explanations. Simulation scoring uses a model to estimate a normalized activation (0-9) for each token in an activating example, given an explanation for the feature. The correlation between the predicted and true activations is the score for the feature. We run simulation scoring on 5 examples per explanation, one from each quantile of cached activations, and filter for features with a simulation score greater than 0.5. We defer to the original work for a more detailed explanation of the method.

To perform simulation scoring, we use Qwen 2.5 7B. We use an all-at-once trick from Bills et al. (2023) to estimate the predicted activations from the top log probabilities for prompt tokens. To filter top explanations, we use Stella 1.5B, Zhang et al. (2025) from the sentence-transformers library Reimers & Gurevych (2019) and choose features with similarity greater than 0.5 with the query. We chose this model for its top ranking classification performance on MTEB (Muennighoff et al., 2023)).

## D  APPENDIX: TRAINING DETAILS

On all tasks, we fine-tune Gemma 2 2B for four epochs with a learning rate of 5e-6 and batch size of 16. We use the adamw optimizer with momentum and weight decay, and default PyTorch configurations (Paszke et al., 2019). We use the NNsight library to perform interventions at each step of training (Fiotto-Kaufman et al., 2024), along with all other intervention experiments we performed.

## E  APPENDIX: AUTOMATED INTERPRETABILITY PERFORMANCE

The performance gap between manual and automatically interpreted features reflects shortcomings in automated interpretability pipelines. It is difficult to design a query at the level of detail with which a human annotator would search through features. For example, an automated explainer in the gender task produces explanations for each feature off the top latents. Querying for "gendered" but not "pronoun" latents is difficult when using a sentence embedding model since the explanations are so similar.

One approach is to provide the explainer with the query and have it provide a score as to how well the information agrees with the query. This has the benefit of not condensing valuable information in the top activations into a single explanation, but scores are not reusable by tasks. Future work could investigate explanations that are more detailed than single, one sentence descriptions and pipelines that incorporate more causal feature information.

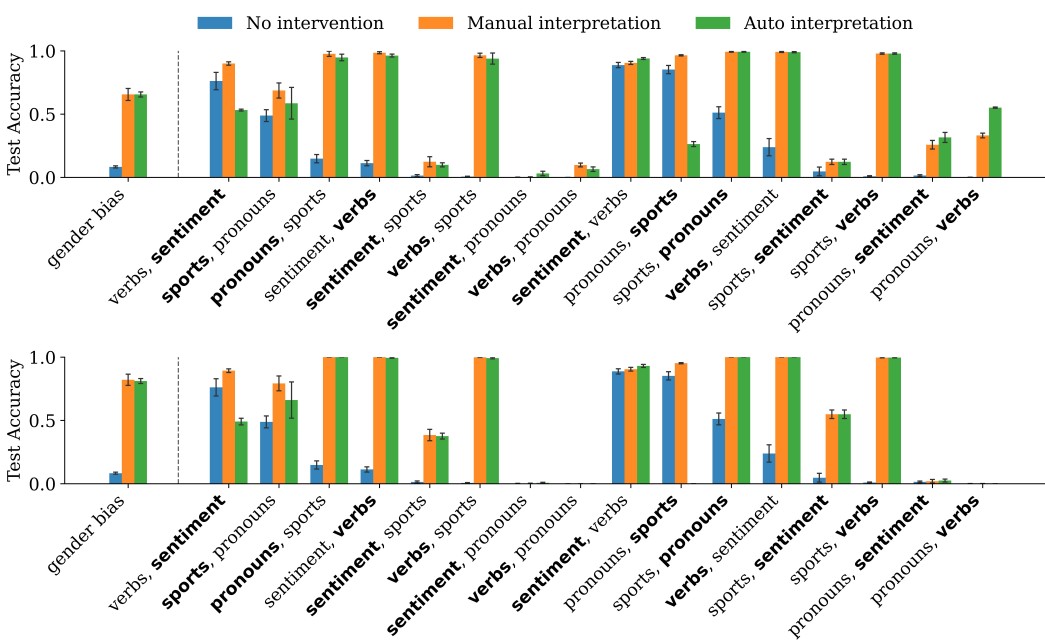

Figure 4: Accuracy on $D_{\text{OOD}}$ for all tasks, ablating latents found using manual interpretation or automatic interpretation. The top plot shows test accuracies and bottom plot shows test accuracies while ablating latents during inference. Using features found by automated interpretability performs about as well as features found by manual inspection.

# F    APPENDIX: BASELINES AND METHOD ABLATIONS

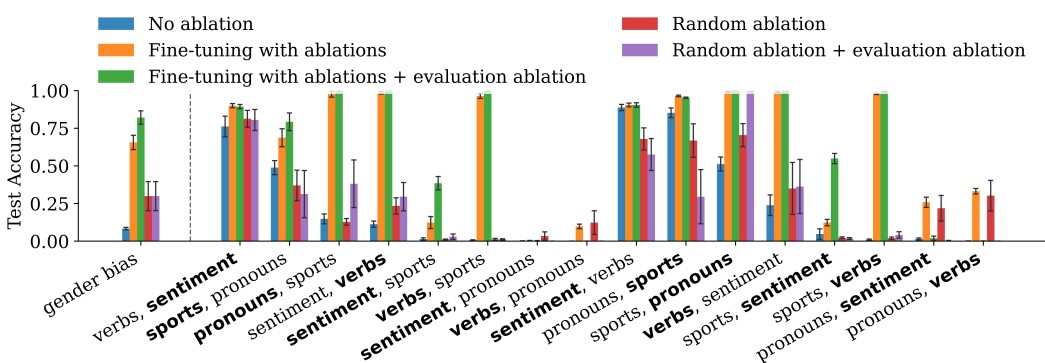

Figure 5: Comparison between ablating latents only during fine-tuning or during fine-tuning and testing, for random and interpreted latents, evaluated on $D_{\text{OOD}}$. Removing the latent ablations during evaluation performs about as well, or just a little bit worse than ablating during evaluation. Random ablations do not work consistently.

Ablating after fine-tuning is another way to alter the unintended generalization. We test two additional methods:

- **Test-only ablation:** we fine-tune the model without interventions and ablate the selected latents only after fine-tuning, when we evaluate performance on $D_{\text{OOD}}$. This shows partial success some of the time but does not perform as well as fine-tuning with ablations.

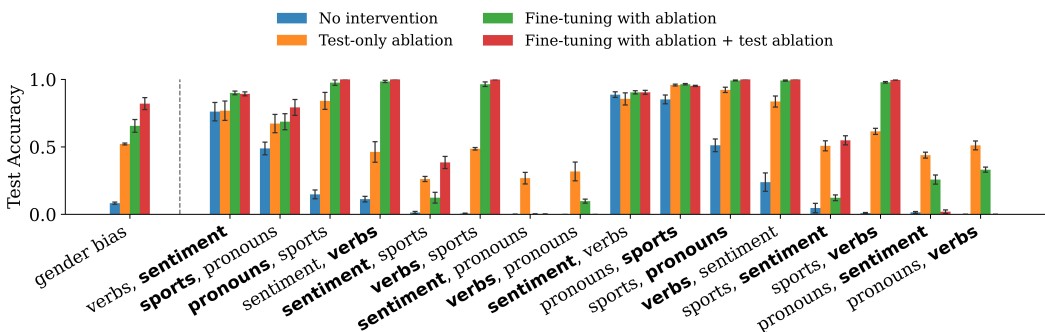

Figure 6: Comparison between ablation during fine-tuning only, during testing only, or during both, evaluated on $D_{\text{OOD}}$. When ablating only during testing, we fine-tune the model on $D_{\text{amb}}$ without interventions and ablate the selected latents when testing on $D_{\text{OOD}}$.

- **Fine-tuning with ablations + test ablations:** we fine-tune with ablations (as described in Section 4) and then ablate during testing too. This method has the highest accuracy overall. However, in some cases the ablations lead to low $D_{\text{val}}$ scores and random guessing.

While ablating during both training and testing achieves the highest accuracy, the small gain in accuracy trades off with the cost of running ablations during inference; each layer with unintended latents must be decomposed, edited, recomposed, and inserted back into the model. Future work could explore methods of slowly turning off ablations during training to close the accuracy gap.

Tables 1 and 2 show skyline performance on the double multiple choice task and the gender bias task, respectively. To measure skyline performance, models were trained on the same distribution as $D_{\text{OOD}}$. These models generalize correctly and achieve at least 95% accuracy on $D_{\text{OOD}}$.

Table 1: Double multiple choice skyline performance

| First Question | Second Question | Mean | Std |
|---|---|---|---|
| verbs | pronouns | 1.00 | 0.000 |
| verbs | sentiment | 0.998 | 0.00293 |
| verbs | sports | 0.999 | 0.00255 |
| pronouns | verbs | 1.00 | 0.000 |
| sentiment | verbs | 1.00 | 0.000 |
| sports | verbs | 1.00 | 0.000 |
| sentiment | verbs | 0.959 | 0.00857 |
| pronouns | verbs | 0.947 | 0.0125 |
| sports | verbs | 0.942 | 0.0202 |
| sports | verbs | 0.993 | 0.00831 |
| pronouns | verbs | 0.991 | 0.00862 |
| sentiment | verbs | 0.998 | 0.00323 |

Table 2: Gender bias skyline performance

| Metric | Mean | Std |
|---|---|---|
| Gender bias | 0.991 | 0.00599 |

