# OpenReview forum: "Steering Fine-Tuning Generalization with Targeted Concept Ablation"
_ICLR.cc/2025/Workshop/BuildingTrust — BuildingTrust_

### Official Review · Reviewer_ywqM · 2025-02-20
**Novel point with potential practical use in real world**

**Rating:** 7
**Confidence:** 3

**Review:**

Strengths:
Clear Motivation & Novel Angle
The authors tackle a critical but underexplored problem: steering the generalization path of a model when multiple solutions exist. By focusing on latent directions associated with undesirable behavior, the paper contributes a unique perspective to the interpretability and controllability of LLMs.

Use of Sparse Autoencoders
The paper capitalizes on recent work in SAEs to decompose model activations into interpretable features. This approach stands out for attempting targeted concept removal, rather than globally retraining on curated data or implementing broad regularization methods.

Well-Designed Synthetic Tasks
Both the gender bias and double multiple-choice tasks are well-defined, enabling a clear demonstration of how a model might rely on spurious correlations. The authors isolate scenarios where the model’s OOD behavior indicates whether it truly learned the intended concept.

Weakness:

Scope of Tasks & Datasets
The tasks, though illustrative, are relatively simple or synthetic. It remains unclear whether the proposed approach would scale effectively to complex real-world domains with more intertwined spurious correlations.

Reliance on SAE Interpretability
Sparse autoencoders, while promising, can exhibit limitations: partial reconstruction errors, unaligned latent spaces, and potential mismatch between a “human concept” and a single latent direction. The paper’s approach hinges on identifying the correct latents to ablate, but the risk remains that some relevant features might be missed or incorrectly labeled.

---

### Official Review · Reviewer_ETdE · 2025-02-28
**SAE-Guided Ablation for Controlled Language Model Generalization**

**Rating:** 6
**Confidence:** 3

**Review:**

**Summary**

This paper presents a novel method that leverages SAEs to steer the generalization behavior of language models during fine-tuning. By identifying and ablating latent features associated with unintended generalizations (such as gender bias and task misalignment in double multiple choice scenarios), the authors aim to guide the model toward the intended behavior. Experiments on a gender bias task and a double multiple choice task show that the method can significantly improve out-of-distribution performance relative to baselines, including random ablations and interventions applied only at test time.

**Strengths**
* **Innovative Methodology**: Combines SAEs with targeted ablation to steer model behavior, offering a novel solution to generalization control.
* **Empirical Validation**: Results on toy tasks—specifically in mitigating spurious correlations in gender bias and improving focus in double multiple choice tasks—demonstrate the potential of the proposed approach.
* **Strong Results**: Demonstrates near-complete elimination of gender bias and significant improvements in 12/16 double-choice scenarios.

**Weaknesses**
* Computational Overhead: The reliance on SAEs and the associated markup for interpreting and ablating features introduces additional compute requirements. The paper does not sufficiently quantify this overhead or discuss its impact on scalability.
* Scalability Concerns: Experiments are limited to a 2B-parameter model and synthetic tasks; applicability to larger models or real-world scenarios is unclear.
* Limited Baselines: There is a noticeable absence of comparisons with alternative methods that do not rely on SAE-based approaches. Such comparisons could clarify whether the performance gains are specific to SAE techniques or are achievable through other means.

**Questions**
* How much does the SAE-based pipeline increase training time compared to standard fine-tuning? Quantifying this would clarify practicality.
* Have you explored alternative methods that might offer similar benefits? A discussion and comparisson on possible non-SAE alternatives would enhance the paper.
* In instances where ablation did not improve performance, could you provide further diagnostics or insights to help understand these shortcomings?

---

### Official Review · Reviewer_bLyh · 2025-03-01
**Great potential for expansion into a longer paper.**

**Rating:** 7
**Confidence:** 3

**Review:**

## Summary
The paper provides a novel technique to steer fine-tuning by ablating unwanted latent 'concepts' using Sparse AutoEnconders. The proposed method performs well on evaluation experiments outperforming a random baseline.

## Strengths

- **Motivation**:  the method is well-motivated and relevant to the field at large
- **Empirical Evaluation**:  the evaluation setup is clear and concise, showing very good and promising results over baseline models and random latent ablations
- **Clarity and Organization**:  the paper is overall quite clear and reads well
- **Impact and Relevance**:  the method is clearly framed within the relevant literature and addresses pressing issues of LLMs like gender biases

## Weaknesses
- **Clarity**: the authors explanation of how they automatically interpret the top 100 activating latents is not too clear to the reader.

## Recommendation
- **Decision**:  **Accept**
- **Key Reasons**:  strong evaluation results, novel methodology, highly relevant to the field

## Additional Feedback
- **Writing and Presentation Suggestions**:  A slightly longer explanation in Appendix B that does not require the reader to read other papers would be quite useful.

---

### Decision · Program_Chairs · 2025-03-01

Accept